# An Overview of siRNA Delivery Strategies for Urological Cancers

**DOI:** 10.3390/pharmaceutics14040718

**Published:** 2022-03-27

**Authors:** Nadia Halib, Nicola Pavan, Carlo Trombetta, Barbara Dapas, Rossella Farra, Bruna Scaggiante, Mario Grassi, Gabriele Grassi

**Affiliations:** 1Department of Basic Sciences & Oral Biology, Faculty of Dentistry, Universiti Sains Islam Malaysia, Kuala Lumpur 55100, Malaysia; nadia.halib@usim.edu.my; 2Urology Clinic, Department of Medical, Surgical and Health Science, University of Trieste, I-34149 Trieste, Italy; nicpavan@gmail.com (N.P.); trombcar@units.it (C.T.); 3Department of Life Sciences, Cattinara University Hospital, Trieste University, Strada di Fiume 447, I-34149 Trieste, Italy; b.dapas@alice.it (B.D.); farrarossella@libero.it (R.F.); bscaggiante@units.it (B.S.); 4Department of Engineering and Architecture, Trieste University, Via Valerio 6, I-34127 Trieste, Italy; mario.grassi@dia.units.it

**Keywords:** bladder cancer, prostate cancer, renal cancer, siRNA, delivery

## Abstract

The treatment of urological cancers has been significantly improved in recent years. However, for the advanced stages of these cancers and/or for those developing resistance, novel therapeutic options need to be developed. Among the innovative strategies, the use of small interfering RNA (siRNA) seems to be of great therapeutic interest. siRNAs are double-stranded RNA molecules which can specifically target virtually any mRNA of pathological genes. For this reason, siRNAs have a great therapeutic potential for human diseases including urological cancers. However, the fragile nature of siRNAs in the biological environment imposes the development of appropriate delivery systems to protect them. Thus, ensuring siRNA reaches its deep tissue target while maintaining structural and functional integrity represents one of the major challenges. To reach this goal, siRNA-based therapies require the development of fine, tailor-made delivery systems. Polymeric nanoparticles, lipid nanoparticles, nanobubbles and magnetic nanoparticles are among nano-delivery systems studied recently to meet this demand. In this review, after an introduction about the main features of urological tumors, we describe siRNA characteristics together with representative delivery systems developed for urology applications; the examples reported are subdivided on the basis of the different delivery materials and on the different urological cancers.

## 1. Introduction

Urological cancers include bladder, prostate and renal cancers [1]. Bladder cancer (BC) is the 11th most commonly diagnosed cancer and has a clear male predominance. Incidence and mortality rates vary across European countries due to differences in risk factors, detection and availability of treatment [2]. Tobacco smoking intensity is the most well-established risk factor, causing 50–65% of male cases and 20–30% of female cases [3]. Nowadays, the most common symptom of BC is painless, visible haematuria, which occurs in about 80–90% of patients [4]. Novel urine biomarker tests outperform cytology and have the potential to improve routine clinical diagnosis and follow-up of BC [5].

Prostate cancer (PC) remains the most common cancer in men in Europe [6]. The incidence of clinically diagnosed PC varies widely and is highest in Northern and Western Europe (>200 per 100,000 men per year) [7]. Nowadays, most patients with PC are diagnosed with early-localized disease and are either asymptomatic or present with lower urinary tract symptoms related to a concomitant benign prostatic hypertrophy [6]. Age, African origin and a family history of PC (affected men of paternal or maternal origin) [8] are well-established risk factors. In addition, a study also suggested that exogenous factors including diet, chronic inflammation, sexual behavior and low exposure to ultraviolet radiation could influence the incidence [9]. About 9% of men with PC have truly hereditary disease, which is associated with an onset 6–7 years earlier than non-hereditary cases and the higher incidence of PC among Africans and Afro-Americans is more aggressive and fatal [10]. Nowadays, trans-rectal ultrasound-guided or trans-perineal ultrasound-guided biopsy using an 18 G needle and a peri-prostatic block is the standard of care for the diagnosis of PC [6]. Unfortunately, 10–20% of all patients will develop castration-resistant PC (crPC) within five years from diagnosis [11]. crPC is defined by disease progression despite androgen deprivation therapy (ADT) and may be manifested by either a continuous increase in levels of serum prostate-specific antigen (PSA), progression of pre-existing disease and/or the appearance of new metastases. crPC is characterized by a poor prognosis and impaired quality of life. In the past, the estimated median survival time of patients with crPC was 9–36 months, depending on the extent of metastasis and symptoms. However, the prognosis has been changed by new hormonal and cytotoxic therapies (e.g., abiraterone, enzalutamide or cabazitaxel). Nowadays, prognosis in crPC is mainly based on retrospective evaluation of completed chemotherapy trials, depending on risk factors [12] or nomograms [13].

The incidence of renal cancer (RC) varies worldwide, being higher in developed countries than in developing countries [14]. Incidence predominates in men, with the male-to-female ratio being 1.5:1, and peaks at age 60–70 years [15]. Established risk factors include smoking tobacco [16], hypertension [17] and obesity [18]. RC can be sporadic or hereditary, but both forms are generally associated with structural alterations of the short arm of chromosome 3 [19]. Currently, the pathological stage, based on tumor size and extent of invasion, is the most important prognostic indicator. From a clinical point of view, an estimated 50% of all RC are discovered incidentally during imaging procedures (abdominal ultrasound [20]) to investigate non-specific abdominal symptoms.

In general, non-metastastic lesions at RC, BC and PCs are treated by minimally invasively surgery [21]. Not every man with RC and PC needs to be treated right away. If the patient has an early-stage PC and RC, many factors such as age and general health, as well as the likelihood of the cancer causing problems, need to be considered before a decision is made. For the advanced forms of RC many different protocols have been developed, including those based on immune checkpoint inhibitor and tyrosine kinase inhibitor [22]. One of the main challenges for these therapeutic options deals with the management of treatment-induced toxicity. In the case of PC, often surgical therapies are not precise and local approaches such as high-intensity focused ultrasound [23], targeted radionuclide therapy [24], hormone therapy and immunotherapy [25] are necessary. Despite the improvements in the therapeutic approaches available, the frequent development of resistance to pharmacological treatments and radiotherapy makes the occurrence of PC relapse an unresolved problem in the field. Additionally, another challenging aspect of PC is the heterogeneity of the disease [26], which indicates that personalized treatments are probably necessary to improve therapeutic outcome. BC is treated with neo-adjuvant or adjuvant chemotherapeutic agents and several other approaches depending on the stage of the disease [27]. However, surgical therapies are sometimes not precise enough and adverse complications including tumor recurrence [28] need to be followed by local ablation using cryotherapy or radiofrequency [28]. Thus, despite a number of therapeutic improvements and the employment of perioperative chemotherapy, the long-term survival rates of patients with BC remained rather unchanged [29].

Based on the above consideration it is clear that novel therapeutic options for urological cancers are required especially for metastatic patients. In this regards, small interfering RNAs (siRNAs), able to down-regulate the expression of the disease-causing gene, represent a novel and promising option. The present review provides a general overview of the novel siRNA-based strategies with potential therapeutic value in the field of urological tumors, i.e., BC, PC and RC. Given the broadness of the field, we focused the attention mainly on recently published papers without, however, omitting some noticeable works of the past 5–10 years. The readers can refer to Ashrafizadeh et al. [30] for an extensive review of the siRNA-based strategies focused on PC. Similarly, for a deeper knowledge about the field of the siRNA-related delivery problems and the many delivery materials available, it is possible to refer to Ashrafizadeh et al. [31], Tatiparti et al. [32], Grassi et al. [33] and Barba et al. [34].

## 2. siRNA Structure, Function and Delivery

siRNA is a double-stranded RNA molecule with 21- and 22-nucleotide generated by ribonuclease III cleavage from longer double-stranded RNA (dsRNAs) [33,35]. After binding to the RNA-induced silencing complex (RISC) in the cytoplasm, the sense strand of siRNA undergoes ejection, while the antisense strand of siRNA targets the complementary messenger RNA (mRNA). Subsequently, partial hybridization of the antisense strand of siRNA with the target mRNA leads to inhibition of translation, while perfect complementary hybridization causes degradation of the mRNA (Figure 1). Thus, siRNA can effectively down-regulate gene expression. Notably, siRNA can be chemically synthesized to target virtually any mRNA of disease-causing gene; thus, they have the potential to be used as a therapeutic agent in many human diseases including urological cancers.

### 2.1. siRNA Delivery Problems

Like all nucleic acid-based molecules [33,36,37], siRNA cannot enter cells on their own and require a delivery agent [38,39,40,41]. When delivered systemically, naked siRNAs have to deal with different biological barriers. In particular, they can: (1) Be degraded by blood nucleases; (2) be eliminated by the phagocytic system, (3) be sequestered by the liver and filtrated by kidney [42] (Figure 2). Additionally, siRNAs have to face the problem of blood wall crossing (extravasation) (4) and migration through the extracellular matrix (5). When they reach the target cell, they cannot efficiently cross the cell membrane (6). Indeed, the presence of phosphate groups in their structure confers to siRNAs a global negative electrical charge that hinders the interaction with the negatively charged surface of the cells, which tends to repulse them. Moreover, siRNA hydrophilic nature prevents the crossing through the hydrophobic inner layer of the cell membrane. Once in the target cell, endosomal escape [43] has to be accomplished (7). If sequestered into endosomes, siRNAs do not have the possibility to get in contact with their targets, thus drastically impairing if not abolishing the biological effect(s). All these obstacles may eventually lead to a negligible effectiveness for siRNA.

Beside the above aspects, another feature of tumor tissue can be considered to optimize siRNA delivery. The aberrant tumor neo-vasculature is responsible for ineffective oxygen delivery in the inner tumor regions. Thus, tumor cells can derive their energy mostly from anaerobic glycolysis, which determines the increased production of lactic acid. This, in concomitance with the reduced H + removal by the defective neo-vasculature, causes the reduction of tumor tissue pH. This feature may be considered (see Section 2.3 and Section 2.4) to generate delivery systems that preferentially release siRNA in low pH [44]. By appropriately taking into account all the above aspects, it is in principle possible to minimize the negative effects of the bio-barriers on siRNA delivery improving the effectiveness of these molecules in the biological environment.

### 2.2. Strategies to Optimize siRNA Delivery

A well-recognized strategy to promote siRNA delivery is based on the use of synthetic vectors that can protect siRNA and potentially deliver it to target cells. Several synthetic materials have been used for delivery of siRNAs as briefly reported below and summarized in Table 1.

### 2.3. Lipid-Based Delivery Materials

The most well known type of lipid-based delivery material is liposomes (Figure 3A). These lipid particles contain an inner aqueous space separated from the outer environment by a bilayer membrane composed of amphiphilic lipids. The hydrophilic heads of the amphiphilic lipids are oriented towards the outer and inner environment; the hydrophobic tails form a hydrophobic environment inside the membrane. A cationic head group characterizes the amphiphilic lipids used for siRNA delivery; this group allows the interaction with the negatively charged phosphate groups present in the siRNA structure.

An interesting group of liposomes are the echogenic liposomes [45]. These are ordinary liposomes, which contain a gas in their internal aqueous environment (Figure 3B). When exposed to ultrasounds consisting of pressure waves with frequencies equal or greater than 20 kHz, echogenic liposomes undergo the phenomenon of cavitation. Cavitation consists of the rapid growth and collapse of bubbles or the sustained oscillatory movement of bubbles. This phenomenon favors the release of siRNA [46] as cavitation causes the formation of transient pores in the cellular membrane that favors the uptake of siRNAs [47].

Recently, a novel class of lipid particles named exosomes has attracted the attention of researchers. These are extracellular nanovesicles with a size of 40–100 nm that are produced by various cells and released into the extracellular environment upon fusion with the plasma membrane. Exosomes arise from early and late endosomes (Figure 3C) that result from invagination of the limited multivesicular body (MVB) membrane [48]. As exosomes can be obtained from patients’ body fluids, they have excellent biodistribution and biocompatibility when reinjected into the same patient for drug delivery purposes [49].

As reported in Section 2.1, the pH responsive liposomes are of interest for the tumor-targeted delivery of siRNAs. Usually they contain a neutral lipid and a weakly acidic amphiphile [50]. In the acidic environment, the negatively charged group of the phospholipid is destabilized, eventually favoring fusion with the cell membrane or with the endosomal membrane and siRNA release. A recent approach to generating pH sensitive liposomes is based on the conjugation with pH-responsive polymers [51].

### 2.4. Polymer-Based Delivery Materials

Polymers are commonly used for siRNA delivery because they are not expensive to produce/isolate and can be easily modified. Typically, they contain cationic groups that allow electrostatic interaction with the negatively charged siRNAs. A widely used polymer for siRNA delivery is polyethylenimine (PEI) [52] that promotes the escape of siRNA from endosome, thereby enhancing the cytosolic availability of siRNAs [53]. Since it displays some toxicity, it is often conjugated with poly(ethylene glycol) (PEG), which has the ability to reduce cytotoxicity and improve stability in the presence of serum proteins [54].

PLGA is a copolymer of polylactic acid (PLA) and polyglycolic acid (PGA) with a variable number of lactic and glycol acid units [55]. It is an FDA-approved polymer that is physically strong and biocompatible/biodegradable.

Chitosan (CH) is obtained by deacetylation of chitin, which is found in the exoskeleton of crustaceans and in the cell walls of fungi. It is a linear polysaccharide with a carbohydrate backbone containing two types of repeating residues, 2-amino-2-deoxy-glucose (glucosamine) and 2-N-acetyl-2-deoxy-glucose (N-glucosamine) [56]. The amino groups give CH a positive charge that enables electrostatic interaction with siRNAs.

Hyaluronic acid (HA) is a linear polysaccharide [57] that, due to the presence of HA receptors in most cancer tissues, has a great potential for targeted drug delivery. In particular, it binds cluster determinant 44 (CD44), an adhesion molecule that is highly expressed in a variety of tumor cells.

Polymers have also been used to create special structures called dendrimers (from the Greek “dendron” (tree) and “meros” (part)). Dendrimers have a central core molecule from which tree-like arms extend in an orderly symmetrical manner [58]. The arms are organized in different layers and the complexity of the dendrimer depends on the number of layers. For siRNA delivery, dendrimers are usually made of polymers that confer a net cationic surface charge to the structure; this allows the siRNA to bind via electrostatic interactions.

As discussed in Section 2.1, the generation of pH responsive polymers can be of great interest for the tumor specific delivery of siRNAs. In this regard, there are two main types of pH responsive polymers [59]: Those with ionizable moieties and those containing acid-labile linkages. In the first case, ionizable moieties such as amines and carboxylic acids are protonated or deprotonated in relation to the pH. The protonation/deprotonation phenomenon determines structural changes in the polymer which allows the release of the drug. In the case of tumor tissue, the decreased pH induces polymer protonation, thus triggering siRNA delivery. Many different polymers can be used for this purpose as long as they contain ionisable moieties. In the second type of pH responsive polymers, the polymer backbone contains acid-labile covalent linkages that are cleaved due to pH decrease. This results in the polymer degradation with the consequent siRNA delivery at the site of increase acidity, i.e, the inner tumor tissue.

### 2.5. Other Delivery Materials

Nucleic acid Aptamers are short single stranded non-coding DNA or RNA molecules [60]. They form secondary/tertiary structures, which eventually determines their 3D shape, responsible for their ability to bind specifically to a large number of target biological molecules. In the biomedical field, aptamers are used as drugs per se [61] or to decorate nanoparticles (of any material) to target specific cellular antigens.

Iron oxide based magnetic nanoparticles (IONPs) are composed of magnetic iron oxides, elements that are widely distributed in nature and easily synthesized in the laboratory. IONPs have a large surface area and can be engineered with functional groups to allow cross-linking with monoclonal antibodies, peptides or small molecules (such as aptamers) for both diagnostic imaging and therapeutic purposes [62]. The major advantage of IONPs in drug delivery is that once the particles enter the blood, they can be recruited to a specific body site by applying an external high-gradient magnetic field.

Carbon nanotubes (CNTs) are an emerging and attractive delivery material for siRNAs [63]. CNTs can be in the form of single-walled carbon nanotubes with a cylindrical shape or in the form of multi-layer graphene sheets wrapped around each other in a cylindrical shape. CNTs are of potential interest for biomedicine and especially for drug/siRNA delivery due to their properties such as large surface area, flexible interaction with cargo, high drug loading capacity and ability to release therapeutic agents at target sites. However, the lack of biodegradability and toxicity has so far limited their full potential in the biomedical field.

## 3. siRNA Delivery in Urological Cancers

Effective use of siRNAs in urological cancers depends on the choice of optimal biological target(s) and appropriate delivery systems. Usually, the choice of biological target of siRNAs depends on exclusive expression in cancer cells, i.e., the siRNA should affect cancer but not normal cells. In most cases, however, such targets are not available. An alternative is to choose targets that are over-expressed in cancer compared to normal tissue. While this strategy is acceptable, it cannot guarantee that normal cells will be unaffected by siRNA action. The ideal delivery system should be able to protect siRNA in the biological environment and, possibly, to allow targeting of cancer cells. Both the choice of the ideal biological target and the optimal delivery system are definitely not trivial tasks. This explains the different attempts found in the literature regarding the choice of optimal siRNA-based approaches for urological cancers. Here, the presented papers are divided according to the type of urological cancer (BC, PC and RC) and to the type of delivery agent (lipid, polymer and “other” delivery approaches) as summarized in Table 2, Table 3 and Table 4.

### 3.1. siRNA Delivery in BC

#### 3.1.1. Lipid-Based Delivery Approaches

Exosomes has been applied to deliver Polo-like kinase-1 (PLK1) siRNA into BC cells in vitro. The PLK1 gene is a key regulator of mitotic progression in mammalian cells and promotes entry into mitosis, spindle formation, sister chromatid segregation and cytokinesis. Overexpression of PLK1 has been demonstrated in many tumor types, including BC, and its expression correlates with higher pathological grade, stage, recurrence and metastasis [64]. PLK1 has also been identified as an independent prognostic marker in non-muscle invasive BC [65], highlighting its importance in tumor progression and the potential for targeted therapies. Moreover, depletion of PLK1 in BC cells has been demonstrated to arrest cell cycle and tumor cell apoptosis [66]. In a recent study [67], exosomes were isolated from embryonic kidney 293 (HEK293) and PLK1 siRNA, or negative control siRNA, was loaded into HEK293 exosomes. After six hours of incubation with BC cell lines UMUC3 and SW780, tumor cells internalized HEK293 exosomes more avidly than normal urothelial cells. In addition, PLK1 siRNA-loaded exosomes significantly decreased the levels of PLK1 mRNA and protein compared to control. While no evidence was provided for the antitumor effects of the PLK1/siRNA/exosome, these data open the possibility of specifically targeting BC cells with exosomes without affecting the corresponding healthy cells. Moreover, a better knowledge of the exosome membrane composition could allow the identification of the targeted cancer molecules by the exosome.

Liposomes, another type of lipid-based carrier material, have been developed to coat siRNA for BC treatment. A cationic liposome (PPCat), consisting of cationic lipid (DOTAP), neutral lipids (cholesterol and DOPE) and a pegylated lipid (DSPE-PEG) was developed to deliver siRNA against survivin [68]. Survivin is highly and selectively expressed in most human cancers, including BC [69], and is an indicator of BC aggressiveness and recurrence [70]. PPCat-siRNA lipoplexes loaded with a non-active control siRNA (siNT) were found to be non-toxic in cultured cells (human bladder transitional RT4 cancer cells). Treatment with PPCat-siSurvivin resulted in significant growth inhibition in the clonogenic assay compared to PCat-siNT. In addition, PPCat-siSurvivin potentiated the activity of mitomycin C (MMC), an antitumor drug. Notably, however, PPCat-siSurvivin did not significantly reduce the in vitro viability of human RT4bladder cancer cells. Consistent with this observation, an in vivo study using a female nude mouse model of BC showed that tumor growth was not affected by PPCat-siSurvivin (intravenous administration). However, PPCat-siSurvivin was able to enhance the MMC activity in vivo. It is possible that the activity of PPCat-siSurvivin can be improved by increasing the dosage. Alternatively, tumor-targeting strategies may improve efficacy by promoting the localization of PPCat-siSurvivin in tumor tissue. Finally, since PPCat-siSurvivin was also not particularly effective in vitro, it may be possible to optimize it with regard to intracellular trafficking (Table 2).

**Table 2 pharmaceutics-14-00718-t002:** siRNA delivery in bladder cancer.

Target mRNA	Delivery System	Disease Model	Reference
PLK-1	exosomes	UMUC3 and SW780 cell line	[67]
Survivin	pegylated lipid	RT4 cell line	[68]
		Mouse model	
Nrf2	dendrimer	T24, 253J B-VC-r and HK-2 cell lines	[71]
Survivin	PLGA-Chitosan	Human ureter model and in vivo mouse bladder; UM-UC-3 cell line and xenograft mouse model	[72]
Bcl2	Chitosan-hyaluronic acid dialdehyde able to bind CD44	T24 cell line, xenograft subcutaneous mouse model	[73]
EIF5A2	Catechin (Mg(II)	T24 cell line, subcutaneous mouse model, Rat in situ model	[74]
RIPK4	halloysite nanotube	in-situ bladder model	[75]
SPAG5	chrysotile nanotubes	T24 cell line,	[76]
		xenograft subcutaneous mouse model, lung metastasis model, in situ rat model	

PLK-1: Polo-like kinase-1; Nrf2: nuclear factor E2-related factor 2; PLGA: copolymer of poly lactic acid and poly glycolic acid; Bcl-2: B-cell lymphoma 2; CD44: cluster determinant 44; EIF5A2: translation initiation factor 5A2; RIPK4: receptor-interacting protein kinase 4; SPAG5: Sperm associated antigen 5.

#### 3.1.2. Polymeric-Based Delivery Approaches

Ambrosio et al. [71] evaluated the biological activity of siRNA targeting nuclear factor E2-related factor 2 (Nrf2). This is a regulator of antioxidant/cytoprotective gene expressions and plays a pivotal role in cancer progression. Moreover, Nrf2 contributes to resistance to pro-oxidant drugs such as cisplatin (CDDP) in various tumors, including BC. Nrf2-siRNA was loaded onto guanidine-terminated carbosilane dendrimer (GCD) by adding the anti-Nrf2 siRNA dropwise to the dendrimer suspension under magnetic stirring. CDDP-resistant BC cells T24 and 253J B-VC-r were used as cellular models. GCD, loaded with a fluorescent dye, was taken up by T24 and 253J B- C-r within 15 min from administration, demonstrating the ability to enter target cells. In both cell lines, Nrf2 expression and activity were significantly down-regulated after administration of Nrf2-siRNA/GCD compared to control. This was paralleled by a significant reduction in CDDP resistance and thus improved CDPP-induced cell death. A slight but significant inhibition of cell proliferation was observed in T24 cells, but not in 253J B-V C-r, after administration of siNrf2-GCD alone. This observation could reflect the different phenotype of the two cell lines. Of note, cells treated with siNrf2-GCD migrated more slowly and showed increased oxidative stress, suggesting that targeting of Nrf2 per se may be of therapeutic value. However, the increase of oxidative stress was quantitatively more pronounced after combined treatment of siNrf2-GCD and CDDP. Finally, the authors showed that the treatment of non-cancerous human kidney HK-2 with siNrf2-GCD and CDDP at the same doses used for BC cells did not result in significant cytotoxicity. This observation raises the possibility that the therapeutic approach developed has a specific anti-BC effect. However, this fact needs to be confirmed in in vivo models of BC.

Among the different polymer used as siRNA delivery agents, PLGA is the most widely used for BC. Martin et al. [72] modified PLGA nanoparticles either by adding chitosan (PLGA-Chitosan) or penetrating polymer via biotin-avidin link (PLGA-AP) (Figure 4A). Chitosan has been used for its mucoadhesive properties, which may promote adhesion to the bladder urothelium and thus prolong the retention of PLAG nanoparticles on the bladder urothelium [77]. AP is an amphipathic, cell-penetrating peptide with short cationic sequences that may facilitate the uptake of nanoparticles by cells via a receptor-independent mechanism. After loading PLGA-Chitosan and PLGA-AP nanoparticles with a fluorescent dye, the authors observed in an ex vivo model (human non-neoplastic dilated ureters) that cell uptake was four and three-fold times higher, respectively, compared to controls. When labelled PLGA-Chitosan and PLGA-AP were instilled into the bladder of mice, fluoresce was observed to increase of about two and nine-fold, respectively, compared to controls, thus qualitatively confirming the ex vivo data. As PLGA-Chitosan nanoparticles were more effectively taken up by the urothelium compared to PLGA-AP, they were considered for further functional testing. Specifically, the authors showed that PLGA-Chitosan particles loaded with an anti-survivin siRNA (siSurvivin) effectively down-regulated survivin expression and significantly reduced the proliferation of the BC cancer cells UM-UC-3. Furthermore, treatment of xenograft tumors with PLGA-Chitosan-siSurvivin resulted in a 65% reduction in tumor volume and a 75% decrease in survivin expression compared to tumor treated controls.

Very recently, Liang et al. [73] developed self-crosslinkable chitosan-hyaluronic acid dialdehyde nanoparticles (CS-HAD) for the targeted delivery of siRNAs to BC cells. Hyaluronic acid, a major component of the extracellular matrix, can bind CD44, which belongs to the cell adhesion molecule family and is highly expressed in a variety of tumor cells [78]; therefore, it is often used as a targeting agent [56]. Moreover, CD44 plays an important role in cellular functions such as cancer cell growth, migration, metastasis and resistance to apoptosis. The author showed that CS-HAD had an increased delivery capacity in BC cells T24 compared to controls. This was dependent on the activation of CD44 after interaction with its ligand HAD. Interestingly, the authors showed that the CS-HAD nanoparticles were able to escape the endosomes. When CS-HAD nanoparticles were loaded with anti B-cell lymphoma 2 (Bcl2) siRNA (CS-HAD-siBcl2), they found a significant reduction of Bcl2 expression, a protein that inhibits apoptosis and promotes oncogenesis. Overexpression of Bcl2 in malignant cells is commonly related to BC initiation, progression and therapy resistance [79]. Finally, in vivo, CS-HAD-siBcl2 inhibited BC growth in a subcutaneous xenograft mouse model without apparent unspecific toxicity.

#### 3.1.3. Other Delivery Approaches

Another method of delivering siRNA in BC consists of the fabrication of nanocomposite particles containing Catechin, a natural anti-cancer compound derived from green tea. Complexes of Mg(II) with Catechin (Mg(II)-Cat) have been used to form nanoparticles containing siRNA targeting translation initiation factor 5A2 (EIF5A2) [74]. EIF5A2 is an oncogene located on chromosome 3q26. It was first discovered in ovarian cancer cells [80], and it is now known that its overexpression predicts poor prognosis in several cancers such as hepatocellular carcinoma [81]. Furthermore, its overexpression correlates with shortened survival in BC patients treated with radical cystectomy [82]. Chen et al. [74] showed that (Mg(II)-Cat nanoparticles) have good biocompatibility and high cellular uptake in the T24 BC cell line. When Mg(II)-Cat was complexed with a siRNA against EIF5A2 (Mg(II)-Cat-siEIF5A2), the resulting nanoparticles were able to significantly knockdown EIF5A2 expression by inhibiting the oncogenic PI3K/Akt signalling pathway, resulting in inhibition of cell proliferation. The PI3K/Akt pathway is a major oncogenic driver that exerts vital functions in cancer growth, survival and progression, and is frequently activated during carcinogenesis. In a subcutaneous xenograft mouse model, the authors showed that Mg(II)-Cat-siEIF5A2 nanoparticles labelled with a fluorescent dye accumulated in the heart, lungs, liver, kidneys, spleen and importantly in the tumor mass following tail vein injection. Remarkably, the naked siEIF5A2 was rapidly cleared from bloodstream, indicating the important role of Mg(II)-Cat in the delivery of siEIF5A2. Finally, in an in situ model of bladder cancer in rats, Mg(II)-Cat-siEIF5A2 reduced tumor size and improved histology as evidenced by tumor lesions being at a lower stage, i.e., better differentiated and more similar to normal tissue.

An interesting technique for the delivery of siRNA deals with the use of nanotubes. Liu et al. [75] demonstrated that natural halloysite nanotubes (HNT) can assist in the delivery of an active siRNA targeting receptor-interacting protein kinase 4 (RIPK4). RIPK4 is a serine/threonine kinase of the RIP kinase family that is highly expressed in BC tissues; it represents an independent prognostic marker for poor survival [83]. The HNTs/siRNA complex increased the serum stability of siRNA, increased its circulation lifetime in blood and promoted cellular uptake and tumor accumulation of siRNA, resulting in a marked down-regulation of RIPK4 expression in an in situ bladder tumor model.

Liu et al. [76] further developed the nanotube delivery approach by the generation of Fe-doped chrysotile nanotubes (FeSiNTs) containing siRNAs directed against sperm-associated antigen 5 (SPAG5) (Figure 4B). SPAG5 is known to exert important functions, including the development and progression of tumor tissues in breast cancer, PC, lung cancer, hepatocellular carcinoma, gastric cancer and cervical cancer. Moreover, upregulation of SPAG5 is associated with poor prognosis in cancer patients [84]. FeSiNTs, with a length of several hundred nanometres, prolonged the half-life of anti SPAG5 siRNA (siSPAG5) in the blood. Moreover, FeSiNTs showed no apparent toxicity when delivered to BC cell T24 at a concentration 20 times higher than that used for effective transfection (200 μg/mL). Using a fluorescent-labelled siRNA (siRNA-FL) delivered by FeSiNTs, the authors found that the optimal siRNA concentration for cellular uptake was 150 nM. Under this condition, the authors proved efficient lysosome escape by FeSiNTs-siRNA, which occurred approximately 4 h after administration to T24 cells. FeSiNTs-siSPAG5 efficiently down-regulated SPAG5 expression resulting in a decrease in cell proliferation and an increase of cancer cell apoptosis. In vivo it demonstrated an excellent tumor tissue penetration ability following intra-tumor injection of FeSiNTs-siRNA-FL. Moreover, when FeSiNTs-siRNA-FL was administered by intravenous injection, the fluorescence intensity into bladder tumor tissue was significantly higher than in the control animal injected with naked siRNA-FL. In this case, the ideal control could have been a different delivery system, as naked siRNA is rapidly degraded in the blood. Therefore, it is not surprising that the concentration in the tumor was lower in the absence of FeSiNTs. Additionally, the accumulation of FeSiNTs-siRNA-FL in the liver and kidney was comparable to that of naked siRNA-FL. This suggests that FeSiNTs has no particular ability to target tumors, although it can promote tumor accumulation by protecting siRNA from degradation in the blood. In a subcutaneous xenograft mouse model of BC, FeSiNTs-siSPAG5 effectively reduced tumor growth and prolonged animal survival compared to controls without causing significant organ toxicity. Notably, in a lung metastatic model, FeSiNTs-siSPAG5 delivered by tail vein injection could effectively reduce lung metastasis, further highlighting its therapeutic potential. Additionally, instillation of FeSiNTs-siSPAG5 in bladder resulted in a significant reduction in the number of tumor lesions in an in situ rat model of BC. Finally, the authors showed that the beneficial effects of FeSiNTs-siSPAG5 depend on the inhibition of the PI3K/AKT/mTOR signaling, an oncogenic pathway that promotes cancer growth, survival and progression.

### 3.2. siRNA Delivery in PC

#### 3.2.1. Lipid-Based Delivery Approaches

LNCaP is an androgen-responsive prostate tumor cell line with a low-aggressive phenotype, while PC3 is a non-androgen-responsive PC cell type characterized by a highly aggressive phenotype [85]. These two cell lines can be chosen to mimic the natural history of PC in the clinic. In the initial stage of PC, the proliferation of PC cells is testosterone driven (androgen responsive), but clonal selection during androgen deprivation therapy may promote the expansion of non-androgen-responsive cells, which may then become predominant. The switch to hormone-refractory PC cells has implications for patient management, as second-line therapy is required. Bae et al. [86] investigated the delivery of an anti-survivin siRNA (siSurvivin) to LNCaP and PC3 (Table 3) using a microbubble-liposome complex (MLC). The MLC were conjugated with anti-human epidermal growth factor receptor type 2 (Her2) antibodies to bind Her2, which is expressed on the surface of the human PC cell line LNCaP and, to a lesser extent, on PC3. Moreover, MLC contained the anti-tumor drug doxorubicin (Dox). In vitro, the authors observed that MLC-siSurvivin-Dox was effectively taken up by LNCaP, but much less by PC3, in agreement with the reduced Her2 expression. Consistent with the different cellular uptake, MLC-siSurvivin-Dox could significantly reduce cell viability in LNCaP but not in PC3. The reduction in viability of LNCaP was further enhanced by the application of ultrasound. It should be reminded that ultrasound can cause MLC collapse, perforations in cell membranes and an increase in the permeability of regional capillaries (when applied in vivo), allowing large molecules to flow into cells. In vivo, no substantial tumor uptake of MLC-siSurvivin-Dox was detectable in aPC3-generated xenograft mouse model of PC. In contrast, in the mouse model generated with LNCaP, uptake was significant especially after the application of ultrasound to the tumor mass. Finally, MLC-siSurvivin-Dox significantly reduced Survivin expression in the LNCaP model.

**Table 3 pharmaceutics-14-00718-t003:** siRNA delivery in prostate cancer.

Target mRNA	Delivery System	Disease Model	Reference
Survivin	Microbubble-liposome	LNCaP, PC3 cell lines, Xenograft mouse model	[86]
CDH2	Commercial liposome	LNCaP cell line	[87]
EphA2	Cationi solid nanoparticles	PC3, DU145 cell lines	[88]
Cldn4	Commercial liposome	LNCaP PC3 cell lines,	[89]
Survivin	PEI conjugated exsosomes	PC3 cell lines	[90]
SIRT6	Exososomes conjugated with A3 aptamer	DU145, PC3, BPH-1 cell lines, xenograft mouse model	[91]
FoxM1	Microbubble-liposome conjugated with A10-3.2 aptamer	LNCaP, PC3 cell lines, xenograft mouse model	[92]
DLX	Commercial liposome	VCaP cell lines,	[93]
Survivin	Poly(propylene) Imine conjugated with an antibody Against PSCA	293TP^SCA/ffluc^ PC3^PSCA^ cell lines, Xenograft mouse model	[94]
Hsp27	Dendrimer like delivery system	PC3 cell lines Xenograft mouse model	[95]
GRP78	CaP-DESP-PEG-RGD	PC3 cell lines Xenograft mouse model	[96]
Metalloproteinase 10	Fe_3_O_4_ nanoparticles conjugated with PEI/PEG	NIH-3T3, PC3 cell lines	[97]
FOAX1	Electroporation	Patients derived organoid	[98]

CDH2: N-cadherin; EphA2: Eph receptor A2; Cldn4: Claudin 4; PEI: polyethylenimine; SIRT6: Mammalian Sirtuins member 6; FoxM1: Forkhead box M1; DLX: Distal-less homeobox; PSCA: prostate stem cell antigen; Hsp27: heat shock protein 27; GRP78: endoplasmic reticulum chaperon; CaP: calcium phosphate; DESP: dioleoylsn-glycerol-3-phosphoethanolamine, dipalmitoyl-glycerol- 3-phosphocholine; PEG: poly(ethylene glycol); RGD: arginine-glycine-aspartic acid; FOAX1: Forkhead box A1.

Lu et al. used a commercial lipid to deliver siRNAs against N-cadherin (CDH2) to PC cells [87]. CDH2, a member of the cadherin family, mediates cancer cell invasion and metastasis; moreover, CDH2 is more expressed in patients with high grade and crPC compared to patients with low grade tumors [99]. The authors used a pool of siRNAs to target CDH2 (siCDH2) in the androgen-responsive LNCaP cells and in the same cell line made resistant to Enzalutamide (LNCaP-R), a drug often used to treat castration-resistant patients. In both cell lines, siCDH2 was able to down-regulate cell viability compared to controls. However, the reduction reached at maximum 25% of control. It is unclear whether this depends on the transfection efficiency (no data were provided in this regard) or on the suboptimal amount of siCDH2 used. A significant reduction in cell migration was also observed; however, the extent of inhibition was very modest in LNCaP-R. Despite being interesting, these data should be confirmed in animal models to understand whether the contained effect reported in vitro can be of relevance in vivo.

Eph receptor A2 (EphA2) belongs to the receptor tyrosine kinase (RTK) family. It regulates cell proliferation, survival and differentiation; moreover, it is overexpressed in many cancers including PC [100]. Oner et al. [88] employed cationic solid lipid nanoparticles (cSLNs) to deliver siRNAs targeting EphA2 (siEphA2). In PC3, DU145 (non-androgen-responsive PC cells) it was shown that cSLNs loaded with a fluorescent siRNA could transfect about 80% of the cells, turning out to be superior to a commercial transfection reagent. Notably, the authors observed that in PC3 the cytoplasmic staining was more homogenous than in DU145 cells. This may suggest that siRNA release from cSLNs particles and/or from endosomes was more efficient in PC3. In line with this hypothesis, cSLNs-siEphA2 complexes were able to effectively down-regulate EphA2 expression in PC3 but not in DU145. Despite the ability to down-regulate expression by cSLNs-siEphA2, no significant reduction in cell viability was observed in PC3 in both 2D or 3D culture systems. Similar results were obtained with regard to PC3 migration. However, cSLNs-siEphA2 potentiated the proliferation inhibition effect of the histone lysine demethylase inhibitor, JIB-04 (5-chloro-N-[(E)-[phenyl(pyridin-2-yl)methylidene] amino]pyridin-2-amine) [101]. This is a novel inhibitor of the histone lysine demethylase (KDM)5A, whose overexpression confers to cancer cells drug resistance. As JIB-04 down-regulates the expression of EphA2, it is possible that the combination with cSLNs-siEphA2 potentiated the down-regulation of EphA2, thus resulting in a significant cell growth inhibition. A corollary of this is that probably cSLNs-siEphA2 alone was not able to sufficiently down-regulate EphA2 to induce PC3 proliferation inhibition. The partial effectiveness in the down-regulation of the expression of EphA2 might not depend on the transfection efficiency that was considerable for cSLNs; thus, it is possible to hypothesize a reduced effectiveness of siEphA2 per se.

Liu et al. employed a commercial lipid to deliver siRNAs against Claudins (Cldns) [89], with transmembrane proteins belonging to a family of tight junction proteins. Among Cldns, the one named Cldn4 is overexpressed in primary and metastatic PC [102]; moreover, Cldn3 is overexpressed in most PC types [103]. Despite no data about the transfection efficiency being provided, the authors showed an excellent expression down-regulation of Cldn3/4 by specific siRNAs (siCldn3 and siCldn4) in LNCaP and PC3 cell lines. This resulted in about 40% cell growth inhibition in LNCaP and 30% in PC3, respectively, for both siClDn3/4. Even more convincing were the results for the inhibition of cell vitality, which reached, in both cell line and for both siClDn3/4, about 60% compared to control. siClDn3/4 effectiveness was further confirmed, showing that the clonogenic ability of LNCaP and PC3 was reduced by about 70% by siClDn3/4. Finally, a significant reduction in cell migration was observed following siClDn3/4 administration to LNCaP and PC3. Notably, the authors observed that the combined administration of siClDn3/4 did not improve the anti-tumor effects observed after the independent administration of each of the two siRNAs. This observation suggests a possible overlapping of functions between ClDn3 and 4 in LNCaP and PC3. Whether this holds true also in other PC cell types remains to be determined. Together these data support the rationale for further investigation in more complex in vivo models of PC of Cldn3/4 targeting by siRNAs.

Recently, Zhupanyn et al. [90] explored the use of PEI-based nanoparticles together with naturally occurring exosomes to combine the beneficial properties of PEI with those of exosomes in siRNA delivery. As a model of PC, the authors considered the PC3 cell line and survivin as siRNA target (siSurv). Survivin is highly expressed in many cancer tissues, and confers on cancer cells the ability to escape apoptosis, thus promoting cell survival [104]. Moreover, survivin promotes mitosis and thus cell proliferation [105]. Therefore, it is an ideal target to down-regulate tumor cell growth. In preliminary tests, the author considered a PC3 cell line expressing the luciferase gene against which a specific siRNA was used (siLuc). The combination of exosomes (isolated from PC3) and PEI-siLuc was far more effective in down-regulating luciferase expression compared to PEI-siLuc alone. A remarkable finding of this work was that the beneficial effect on siLuc was substantially independent from the source of exosome. In particular, exosomes isolated from other cell types (ovarian or bone cancer cell lines) gave comparable results to exosomes isolated from PC3 in promoting siLuc effects. This observation argues against a targeting role of exosome into the cells, which originated them. In a xenograft subcutaneous mouse model of PC, it was than shown that following i.v. injection of exosomes/PEI-siSurv (10 µg in three consecutive administrations) tumor mass was inhibited for about 45%. Moreover, 50% down-regulation of survivin expression in tumor mass was reported compared to controls.

An attractive alternative to the use of targeted antibody/peptide to drive siRNA to cancer cells is of the use of aptamer. In this regard, exosomes were recently conjugated with the aptamer E3 (AptE3) [91]. This is an RNA aptamer, 36 nucleotides long that was generated to be specifically internalized into PC cells. Notably, AptE3 is not internalized into normal prostate cells, thus providing a remarkable specificity of action [106]. The surface of exosomes was modified by binding AptE3 linked to maleimide-PEG-cholesterol molecules (Figure 5A). The targeted exosomes were loaded with a siRNA directed against the mRNA of the Mammalian Sirtuins member 6 (SIRT6). SIRT6 is overexpressed in PC tissue and has an oncogenic role in PC progression [107]. As an in vitro model of PC cells, DU145 cells and PC3 were used, both being non-androgen-responsive PC cells characterized by a moderate and high aggressive phenotype, respectively [85]. Using a fluorescent-labelled siRNA carried by the AptE3-modified exosomes, the authors demonstrated an efficient uptake compared to control non-cancerous epithelial BPH-1 cells. In a subcutaneous mouse model of PC (DU145 were used for grafting), AptE3-modified exosomes carrying a bioluminescent siRNA were injected intravenously into the mouse. The authors showed a strong accumulation of bioluminescence in the tumor region but also in the liver. The latter is not surprising, as the liver is one of the main sites of clearing particle from the blood. Unfortunately, a control consisting of exosome conjugated to non-active Apt was not tested, so it is only partially possible to assess the specificity of the system in vivo. In agreement with the uptake results, the intravenous delivery of a siRNA against SIRT6 resulted in a significant reduction of tumor mass growth. Finally, systemic delivery of the AptE3-modified exosomes reduced orthotopic tumor growth and reduced liver metastasis in an orthotopic PC model (PC3 were used for grafting). Despite the uncertainty about the specificity of AptE3-modified exosomes, this represents a promising approach to delivery of siRNAs to PC cells.

The use of targeting aptamer has also been applied to microbubble liposome. Wu et al. [92] developed an interesting targeted ultrasound (US)-sensitive nanobubble by dissolving fixed ratios of 1,2-Dipalmitoyl-sn-glycero-3-phosphocholine (DPPC), 1,2-distearoyl-sn-glycero-3-phosphoethanolamine (DSPE-PEG2000-COOH) and 3b-[N-(N’,N’-dimethylaminoethane)-carbamoyl]- cholesterol hydrochloride (DC-cholesterol) in chloroform. The cationic nanobubbles (CNBs) produced were later conjugated with an A10-3.2 aptamer. This RNA aptamer is designed to target the prostate-specific membrane antigen (PSMA) [108]. PSMA is a transmembrane protein that is typically upregulated in androgen-dependent PC cells such as LNCaP, but much less in androgen-independent ones such as PC3. CNBs containing A10-3.2 aptamer were loaded with a siRNA targeting Forkhead box M1 (siFoxM1). FoxM1, a gene involved in proliferation, is overexpressed in PC [109] and its inhibition down-regulates cell proliferation and promotes apoptosis. The average size of the CNBs-A10-3.2 was 479.83 ± 24.5 nm, with a poly dispersity index of 0.178 ± 0.023. CNBs-A10-3.2 loaded with a fluorescent dye demonstrated high specificity in binding to LNCaP cells overexpressing PSMA. In contrast, no specific binding was observed in PSMA-negative PC3 cells. This observation indicates that the efficacy of the developed delivery approach is limited to PSMA-positive PC cells and therefore some patients may not benefit from this approach. In vitro in LNCaP, siFoxM1-CNBs-A10-3.2 resulted in a significant reduction in FoxM1 expression compared to treated control cells. Moreover, a stronger inhibition was observed compared to cells treated with siFoxM1-CNBs, i.e., nanoparticles lacking the targeting aptamer A10-3.2. This proves the targeting ability of A10-3.2. In LNCaP, but not in PC3, siFoxM1-CNBs-A10-3.2 combined with ultrasound-mediated nanobubble destruction significantly increased the number of cells in G1 phase and reduced that of S phase cells. Moreover, cell apoptosis was increased. In vivo, in a xenograft mouse model of PC, siFoxM1-CNBs-A10-3.2, but far less siFoxM1-CNBs, accumulated at the tumor site. In line with this observation, siFoxM1-CNBs-A10-3.2 led to a more evident inhibition of tumor growth and extension of animal survival of the mice, compared to siFoxM1-CNBs. Finally, only modest toxicity of siFoxM1-CNBs-A10-3.2 was detected. Taken together these data support the potential validity of the developed approach, which is, however, limited to the PC forms that overexpress PSMA.

The distal-less homeobox (DLX) genes belong to the homeobox containing family of transcription factors (TFs). Its deregulation has been related to different human tumors, including PC [110]; moreover, it represents an established biomarker for PC [111]. Recently, Goel et al. [93] explored the consequences of DLX1 silencing by siRNAs in VCaP, an androgen responsive PC cell line [112]. siRNAs directed against DLX1 (siDLX1) were delivered to cultured VCaP by a commercial liposome at time “0” and twenty four hours thereafter. Following the double transfection procedure, DLX1 silencing resulted in cell cycle arrest characterized by a certain increase of G1/G0 cells and a slight decrease of G2/M cells; no significant variation of the S phase cells was evident compared to control siRNA. Additionally, DLX1 silencing promoted cell apoptosis, further contributing to the inhibition of VCaP expansion. These observations together with the fact that the increased expression of DLX1 occurred in about 60% of the patients analyzed by the authors strongly support the appropriateness of DLX1 targeting for novel anti PC approaches.

#### 3.2.2. Polymer-Based Delivery Approaches

Jugel et al. [94] have developed an interesting polyplex that offers the possibility of carrying a therapeutic siRNA against survivin (siSurv) and targeting the complex to the surface marker prostate stem cell antigen (PSCA). PSCA is an associated glycophosphatidylinositol (GPI)-anchored cell surface antigen that is overexpressed in PC cells, including high-grade prostatic intraepithelial neoplasms and androgen-dependent/independent tumors [113]. This feature makes PSCA an attractive candidate for targeted drug delivery strategies in PC. The polyplex consisted of two main components. One contained maltose-modified poly(propylene imine) polymers capable of binding siSurv (siSurv-PI); this part was also conjugated with biotin. An antibody able to target PSCA formed the second part (Ig-PSCA); in this case, a biotin-binding moiety was also added. The two parts were combined with a multi-biotin binding element called neutravidin (Figure 5B). For in vitro studies, the authors used the human embryonic kidney cell line 293TPSCA which overexpresses PSCA. While not a PC cell line, these cells can be used to study the targeting and efficacy of the polyplex produced. Three different polypexes were tested: The first contained siSurv-PI/Ig-PSCA, the second siSurv-PI/Ig-control (PSCA not binding) and the third just siSurv-PI. In all cases, the siSurv was conjugated with a red fluorescent dye to allow the detection of siSurv in the cells. While siSurv-PI/ Ig-control and siSurv-PI gave no or minimal cell fluorescence, siSurv-PI/Ig-PSCA clearly stained most of the cells, supporting its targeting specificity. Notably, the PI/Ig-PSCA polyplex loaded with an anti-luciferase siRNA (siLuc) to form siLuc-PI/Ig-PSCA was effective in reducing luciferase expression (compared to control) in 293T^PSCA/ffLuc^ overexpressing the luciferase gene. This suggests the ability of siLuc-PI/Ig-PSCA to functionally deliver the carried siRNA. These data were also reproduced in the more suitable model of the PC cell line PC3^PSCA^. PC3 are non-androgen-responsive PC cells that exhibit a very aggressive phenotype [85,114]. In vivo, in a xenograft mouse model of PC, siSurv-PI/Ig-PSCA and the control polyplexes were injected intraperitoneally (i.p.) every third day for 17 days into PC3^PSCA^ tumor-bearing mice. siSurv-PI/Ig-PSCA was most effective in down-regulating tumor cell growth. However, also siLuc-PI/Ig-PSCA and siSurv-PI could reduce tumor growth, albeit to a much lesser extent. Since siLuc has no anti-tumor activity, in the first case, it is possible only the targeting PSCA affects tumor cell growth. This suggests that the mechanisms of action of siSurv-PI/Ig-PSCA are mediated by both siSurv and PSCA targeting. In the second case, it cannot be ruled out that siSurv-PI interacts non-specifically with tumor cells, releasing the active siSurv.

Dong et al. [95] developed a dendrimer-like delivery system for siRNA based on a backbone of DSPE (natural phospholipid derivatives 1,2-distearoyl-sn-glycero-3-phosphoethanolamine) linked to different generation of dendritic L-lysine. DSPE constitutes the hydrophobic tail, while L-lysine is the hydrophobic tail. Two dendrimers were generated, one with a reduced degree of L-lysine dendrome residues (DSPE-KK2) and one with an increased degree of L-lysine dendrome residues (DSPE-KK2K4). The generated dendrimers were able to compact siRNA, thus providing protection in the biological environment. As the siRNA target, the heat shock protein 27 (Hsp27) was chosen—an ATP-independent molecular chaperone able to favor prostate tumor cell survival [115]. In PC3, siHsp27/DSPE-KK2 was better internalized than siHsp27/DSPE-KK2K4, resulting in a homogeneous staining of the cytoplasm. This allowed the authors to conclude that probably siHsp27/DSPE-KK2 promoted endosome escape more efficiently than siHsp27/DSPE-KK2K4. siHsp27/DSPE-KK2 effectively down-regulated the expression of Hsp27. Notably, no major signs of unspecific cytotoxicity were observed as evaluated by measuring the levels of the necrotic marker LDH. Moreover, siHsp27/DSPE-KK2 induced a significant reduction in PC3 proliferation and migration/invasion. In a xenograft subcutaneous mouse model of PC, siHsp27/DSPE-KK2 was injected intratumorally twice per week for three weeks. At the end of the treatment cycle, siHsp27/DSPE-KK2 significantly down-regulated Hsp27 expression in the tumor, slowing down tumor growth as also confirmed by the reduced levels of Ki67, a known marker of cell proliferation. The lack of significant unspecific toxicity was confirmed in vivo by measuring the levels of the liver function markers ALT/AST and renal function urea. Further tests in vivo following systemic delivery of siHsp27/DSPE-KK2 will fully determine the potential therapeutic value of this interesting approach.

Zhang et al. [96] developed a complex delivery system for siRNA based on a calcium phosphate (CaP) core. The positive electric charge of Ca^2+^ can efficiently bind the negatively charged siRNA forming the CaP/siRNA core. To increase the nanoparticle stability, the CaP/siRNA core was linked with the lipids dioleoylsn-glycerol-3-phosphoethanolamine, dipalmitoyl-glycerol- 3-phosphocholine (DESP). The lipid component was also used to allow the encapsulation of the highly lipophilic drug docetaxel (DTXL), used in the treatment of metastatic CRPC patients [116]. Finally, DESP was linked to the arginine-glycine-aspartic acid (RGD) moiety via PEG (CaP-DESP-PEG-RGD). RGD has the ability to target defined integrins expressed on the endothelial cells of tumor neo-vasculature but not on endothelial cells of normal vessels [117]. Thus, the delivery system prepared has the ability to bind siRNA (via CaP), DTXL (via DESP), to target tumor neo-vasculature (via RGD), to be protected from endocytosis by phagocytes and to be stable in the blood (via PEG). As a target for siRNA, the authors chose GRP78, an endoplasmic reticulum chaperon, involved in protein folding and assembly and protein quality control. GRP78 is involved in the drug resistance process in PC and is an effector protein of the androgen receptor [118]. In PC3, CaP-DESP-PEG-RGD-siGRP78 effectively down-regulated GRP78 expression compared to control. With regard to the effect on cell viability reduction, it was shown that CaP-DESP-PEG-RGD-siGRP78-DTXL was more effective than free DTXL + siRNA and free DTXL. This proves the positive role of the delivery complex. However, as the effect of CaP-DESP-PEG-RGD-siGRP78 and CaP-DESP-PEG-RGD-DTXL alone was not evaluated, it is not possible to state whether siGRP87 cooperated with DTXL to down-regulate cell viability. The authors also showed increased apoptosis rate for CaP-DESP-PEG-RGD-siGRP78-DTXL compared to control. It would have been interesting to test the delivery system in cells that do not express the RGD receptor to define the targeting effectiveness of the delivery complex. In a xenograft subcutaneous mice model of PC (generated using human PC3), CaP-DESP-PEG-RGD-siGRP78-DTXL was delivered systemically every three days for 24 days. At the end of the treatment, tumor mass growth was effectively reduced compared to free DTXL, free siRNA and free DTXL + siRNA. This observation confirms the effectiveness of the delivery system developed in vivo but, once again, does not allow stating whether siRNA cooperates with DTXL when delivered by DESP-PEG-RGD. Finally, the unspecific toxicity of the empty CaP-DESP-PEG-RGD was negligible in vivo as well as in vitro, supporting the appropriateness of the delivery system developed.

#### 3.2.3. Other Delivery Approaches

The utilization of magnetic nanoparticles is commonly adopted for siRNA delivery systems. Recently, Fe_3_O_4_ nanoparticles have been developed to deliver siRNA targeting metalloproteinase 10 (ADAM10) [97] in PC cells. ADAM10 is a secreted metalloproteinase involved in various physiological processes and in tumorigenesis. Its function is essentially related to the ability to degrade different extracellular matrix proteins and activate other metalloproteases. The upregulation of ADAM10 has been observed in various tumor types including liver adenocarcinoma, oral squamous cell carcinoma and PC [119]. Moreover, the development of PC was found to be dependent on the nuclear translocation of ADAM10, whereas in benign prostatic hypertrophy it is mostly bound to the cell membrane. In order to improve the biocompatibility, the citrate-stabilized magnetic nanoparticles were coated with PEI and PEG. The PEI/PEG coating made it possible to obtain nanoparticles with superparamagnetic iron oxide nanoparticles (SPIONs) in the core. It should be remembered that PEI favors the dispersion and stability of the nanoparticles and offers the possibility of conjugating siRNA via electrostatic interaction. Moreover, PEG increases biocompatibility, as shown by the fact that excellent biocompatibility was observed in mouse fibroblast NIH 3T3 and in PC3 cells. The size of the prepared nanoparticles was 15.82 (±9.07) nm, while the hydrodynamic size was about 79.20 (±0.68) nm. The PEG-PEI–Fe_3_O_4_ SPIONs loaded with the anti ADAM10 siRNAs were successfully delivered to PC3 cells resulting in a significant decrease in cell viability. Confocal microscopy confirmed that siRNA-loaded nanoparticles entered the cytosol of tumor cells. While these data are promising, they need to be confirmed in the animal model of PC.

In contrast to the approaches based on the use of delivery materials, the technique known as electroporation just needs the generation of an electrical field to be applied to cells. The electric pulse temporarily alters the cell membrane, increasing its permeability and thus allowing therapeutic molecules, including siRNAs, to be internalized into the cells. This technique, firstly used in vivo in a gene in 1991 [120], is nowadays used in many preclinical and clinical studies (https://clinicaltrials.gov/ access date 15 March 2022). This delivery approach was recently employed to study the role of Forkhead box A1 (FOXA1) in PC [98]. FOXA1 is a protein able to induce open chromatin conformation, thus allowing other transcription factors to bind to their DNA target sites [121]. In prostate cells, FOXA1 interacts with the Androgen Receptor (AR), helping it to promote the growth and survival of normal prostate. In PC, FOXA1 can drive tumor onset and progression [122] being able to reprogram AR binding sites, thus driving oncogenic programs. In particular, FOXA1 plays a relevant role in the generation of neuroendocrine prostate cancer (NEPC). This is an aggressive form of PC, which pathologically expresses genes typical of neuroendocrine tissues. Together, the above considerations indicate that FOAX1 targeting may be of therapeutic potential for PC and NEPC in particular. Baca et al. [98] deepen our knowledge about FOAX1 in NEPC, employing different elegant approaches including FOAX1 silencing by siRNA. In patient-derived organoids of NEPC (WCM154) [123] a pool of anti FOAX1 siRNA (siFOAX1) was delivered to WCM154 by electroporation. Because of this test, the authors could show that siFOAX1 was able to reduce the cell growth of about 50% of control. This and the other experimental approaches undertaken by the authors provide the rationale for targeting FOAX1 in future treatments for NEPC.

### 3.3. siRNA Delivery in RC

#### 3.3.1. Lipid-Based Delivery Approaches

Sakurai et al. [124] developed a liposomal siRNA delivery system called MEND: Multifunctional envelope-type nanodevice (Table 4). It contains, among others, a cationic lipid, YSK05, with high membrane fusogenic properties. In particular, YSK05 contains positively charged residues that facilitate interaction with the endosomal membrane. This allows the fusion of MEND with the exosome membrane, leading to an efficient release of siRNA from the endosome (endosome escape). To down-regulate the expression of PLK1, which is involved in the genesis of different human tumors, the MEND nanoparticles were loaded with an anti-PLK1 siRNA (siPLK1). In a subcutaneous mouse xenograft model of RC with OS-RC-2 (renal cell carcinoma cells), the MEND-siPLK1 nanoparticles effectively reduced PLK1 mRNA levels compared to the control. Nevertheless, MEND-siPLK1 failed to inhibit tumor growth. This observation could be due to suboptimal down-regulation of PLK1, but also to biological reasons, such as a certain resistance of OS-RC-2 to PLK1 down-regulation. To try to overcome the poor efficacy of MEND-siPLK1, combined delivery with the antitumor drug doxorubicin (DOX) was explored. For optimal delivery, DOX was formulated in liposomes (DOX-lip). MEND-siPLK1 in combination with DOX-lip resulted in a significant reduction in tumor growth without any evidence of acute toxicity. Importantly, administration of a non-functional siRNA (MEND-sLuc) did not confer antitumor effect to the DOX-lip complex. This supports the functional role of siPLK1. The authors explained the potentiating effect of MEND-siPLK1 on DOX-lip as follows. Silencing of PLK1 was responsible for the accumulation of cells in the G2 phase of the cell cycle. Since cells blocked in the G2 phase are more sensitive to DOX, it is conceivable that this block was responsible for the increased DOX activity in OS-RC-2.

**Table 4 pharmaceutics-14-00718-t004:** siRNA delivery in renal cancer.

Target mRNA	Delivery System	Disease Model	Reference
PLK1	Liposome containing the cationic lipid YSK05	OS-RC-2 cell line, Xenograft mouse model	[124]
HMGA2	Commercial liposome	ACHN cell line	[125]
KSP/VEGF	Lipidi particle	Clinical trial	[126,127]
Lim-1	Polydiacetylenic nanofibers	786-O, cell line, xenograft mouse model	[128]
Survival gene	Glycogen	MDA-MB-231-luc2 HK2 cell lines	[129]

PLK-1: Polo-like kinase-1; HMGA2: High mobility group AT-hook 2; KSP: kinesin spindle protein; VEGF: vascular endothelial growth factor.

High mobility group AT-hook 2 (HMGA2) is a non-histone protein that binds chromatin, modifying its structure by bending and stretching; this in turn modifies the access of the transcription factor to the gene promoter [130]. Thus, HMGA2 indirectly regulates gene expression. HMGA2, which regulates cell cycle and apoptosis, is upregulated in RC [131]. Very recently, Chen et al. [125] employed a commercial liposome to deliver three anti-HMGA2 siRNAs (siHMGA2s) to the human renal carcinoma cell ACHN. After having shown the effectiveness in reducing HMGA2 expression (down to 20% mRNA of control), the authors observed an increase in G1/G0 cells and a decrease of S/G2-M cells. This was due to a decrease of the G1 cyclin D1 and of E2F1, both promoters of the G1/G0 to S phase transition [132]. The authors also reported the induction of apoptosis, which was likely due to the decrease of the levels of BcL2, a promoter of apoptosis. Thus, the combined reduction in cell growth and apoptosis induction was responsible for the down-regulation of ACHN expansion. Despite the authors also employing in the work the normal human renal proximal tubular epithelial cell line (HKC), the effects of HMGA2 silencing was not studied in HKC. This test would have been useful to determine the potential side effects on healthy kidney tissue. Finally, experiments in the relevant in vivo model of RC are necessary to better determine the effectiveness of HMAG2 silencing.

In 2009, Alnylam Pharmaceuticals, in collaboration with Tekmira, encapsulated siRNAs in stable nucleic acid-lipid particles (ALN-VSP02). ALN-VSP02 contains two siRNAs that target kinesin spindle protein (KSP) and vascular endothelial growth factor (VEGF) mRNAs. KSP, a member of the kinesin superfamily of microtubule-based motors, is a key regulator of mitosis and its inhibition leads to cell cycle arrest in mitosis. VEGF is a growth factor that promotes both vasculogenesis (de novo vessel formation) and angiogenesis (blood vessel formation from pre-existing vasculature) and therefore plays an important role in the development of highly vascularized tumors. Both KSP and VEGF are overexpressed in cancer patients, and down-regulation of KSP/VEGF results in inhibition of tumor cell proliferation and angiogenesis. ALN-VSP02 entered into phase I trial, which included the dose-escalation phase and expansion phase [126,127]. ALN-VSP02 particles are 80–100 nm in diameter and are essentially uncharged particles. When administered systemically, ALN-VSP02 localize in the liver and spleen. This is most likely due to the highly fenestrated endothelium in these organs, which favors passive extravasation. For the same reason, ALN-VSP02 tends to accumulate in tumor tissues that have a leaky microvasculature. In the dose-escalation phase, 30 patients with hepatic and/or extrahepatic tumors were enrolled for intravenous administration of ALN-VSP02 at a dose of 0.1 to 1.5 mg/kg. Three patients with renal cell cancer or pancreatic neuroendocrine tumor experienced 12–18 months of tumor stabilization. Further data in a larger patient cohort are needed to fully understand the relevance of this therapeutic approach.

#### 3.3.2. Polymer-Based Delivery Approaches

Neuberg et al. [128] generated polydiacetylenic nanofibers by photopolymerization to optimize intracellular delivery of siRNAs. Diacetylenic surfactants consist of a long C25 hydrocarbon chain and are capable of self-assembling into many supramolecular structures. Notably, the diacetylenic (DA) systems can be polymerized into the so-called PDA (PolyDiAcetylenic) upon UV irradiation. After polymerization, the PDA selected by the authors formed small fibers, termed PDA-nanofibers (PDA-Nfs). In a first set of experiments, the authors tested PDA-Nfs loaded with an anti-luciferase siRNA (siLuc, PDA-Nfs-siLuc) in the human RC cell line 786-O which express the luciferase gene, showing a high degree of luciferase silencing. The ability of PDA-Nfs to silence the oncogene Lim-1 loading an anti-Lim-1 siRNA (siLim-1) was then tested. Lim-1 is a transcription factor required for normal organogenesis, including nephrogenesis, and regulates cell movement, differentiation and growth. Its expression is reactivated in RC and it is involved in tumor cell growth and survival [133]. The authors showed that PDA-Nfs/siLim-1 complexes successfully down-regulated Lim-1 expression in the 786-O cell line. It would have been interesting to obtain data on the phenotypic effect, i.e., the extent of cell death induction. This in vitro observation was confirmed in subcutaneous tumor xenografts obtained grafting 786-O cells in nude mice after intraperitoneal injection of PDA-Nfs-siLim-1. The systemic route chosen by the authors suggests that PDA-Nfs/siLim-1 can effectively reach the target mRNA; however, data on its ability to reduce tumor growth would have been interesting in this case as well.

Due to their biodegradability, biocompatibility, low immunogenicity and cytotoxicity, the polysaccharide-based sort are often considered for the development of siRNA delivery systems [134]. Among these [56], glycogen is considered an attractive molecule when properly functionalized. In this regard, very recently, Racaniello et al. [129] investigated the use of a glycogen derivative (PG) extracted from mussels and composed of D-glucose molecules linked by α (1 → 4) bonds with branches every 5–10 glucose units, linked by α (1 → 6) bonds. PG, which has a highly branched structure, was processed to a dendrimeric structure. Moreover, PG has been functionalized with N,N-dialkylamino alkyl halides containing two amino groups. The first was to ensure the binding with nucleic acids through electrostatic interaction; the second was to promote endosomal escape by proton sponge effect. The authors were able to show an efficient siRNA retention of the functionalized PG (PGF). In vitro, PGF loaded with a model siRNA (PGF-siRNA) was efficient in releasing siRNA into target cells, as demonstrated by confocal microscopy. To assess cytotoxicity, an MTT assay was performed on the breast cancer cell line MDA-MB-231-luc2. The values obtained demonstrated that PGF-siRNA was not cytotoxic, even at higher concentration. The transfection efficacy of PGF-siRNA was evaluated using a siRNA targeting ubiquitously expressed human genes essential to cell survival. The PGF-siRNA complexes were tested on the HK2 cell, an immortalized primary tubular epithelial cell line derived from normal adult human kidney. The siRNA chosen was a cell death siRNA targeting ubiquitously expressed human genes essential to cell survival. In this experiment, the researchers showed that PGF-siRNA was able to induce HK2 cell death. While these data support the efficacy of PGF-siRNA in RC, it would have been interesting to have data on uptake efficiencies in HK2 to allow parallelism between PGF-siRNA efficacy and uptake level.

## 4. Conclusions

Without a suitable delivery system, siRNAs have negligible effects in the biological environment of urological cancers as well as in other forms of cancers. Thus, optimized delivery materials need to be developed. In this regard, we believe that at least three different aspects should be considered. The first aspect deals with the ability to properly release siRNAs [135] and to protect them from degradation. This goal does not seem to be excessively tricky, as shown by the number of works published in many field of human pathologies including urological cancers. However, siRNA protection against degradation is not enough for an optimized therapeutic approach. Thus, the second aspect to be considered consists of the development of delivery materials able to target cancer cells without affecting the normal counterpart cells. This is particularly important when systemic delivery is required, like in the case of urological cancers. A targeted system not only can preserve the function of the normal tissue where the tumor developed, it can also preserve the function of distal tissues where metastasis can occur. The choice of the targeting molecules (chemical targeting) on the tumor cells is, however, not trivial. Indeed, tumor cells may change their phenotypes over time, thus becoming “resistant” to a defined targeting strategy. Moreover, often, cancer cells share similar antigens with the normal counterpart. Thus, an accurate search for cancer specific urological markers should be undertaken. The so-called physical targeting approaches (magnetic particles, echogenic liposome, pH responsive materials and particle size/shape) may represent alternatives, although with some known limitations. Obviously, the physical strategy may be used in combination with the chemical targeting. The third aspect to be considered for optimal delivery deals with the evaluation of the biological features of tumor cells. In particular, the possibility to target the mRNA of a pathologic gene(s) in the cancer cell not expressed by the normal tissue counterpart should guarantee high specificity of action. However, the identification of such targets is not an easy task, as often they are also expressed in normal cells, although to a lower extent. Finally, we believe that future approaches should consider the development of delivery systems with a double level of specificity: The first coming from the use of a material able to target cancer cells combined with the use of siRNA directed against the mRNA of pathological genes overexpressed in cancer cells and not expressed/poorly expressed in normal counterpart cells.

In conclusion, we believe that with the proper consideration of the issues above reported and in the light of the promising works described in this review, the future of siRNAs as potential therapeutic molecules in urological treatment may be realized.

## Figures and Tables

**Figure 1 pharmaceutics-14-00718-f001:**
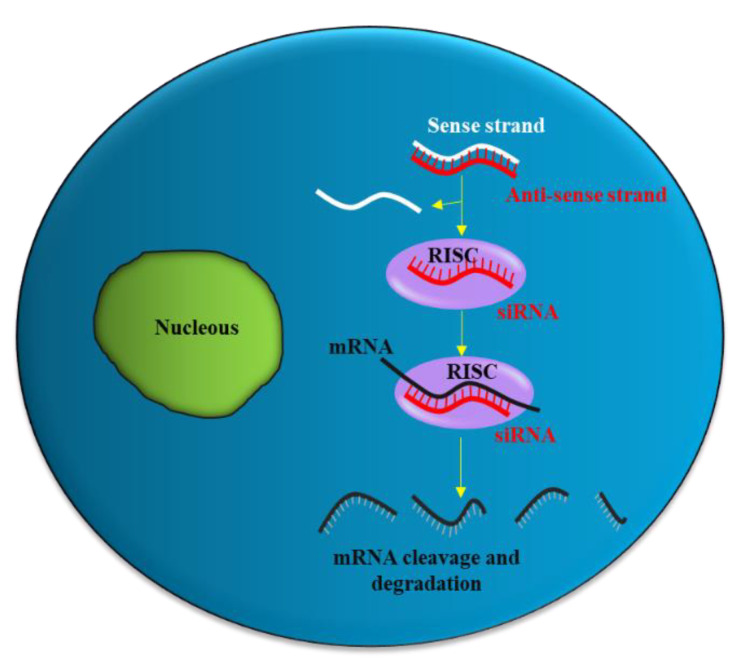
siRNA mechanism of action. The antisense strand (red) of the siRNA is taken up by a catalytic protein complex (RNA-induced silencing complex, RISC), and the sense strand (white) of the siRNA is discarded. The antisense strand drives RISC to a target mRNA (black), which results in specific, RISC-mediated mRNA degradation.

**Figure 2 pharmaceutics-14-00718-f002:**
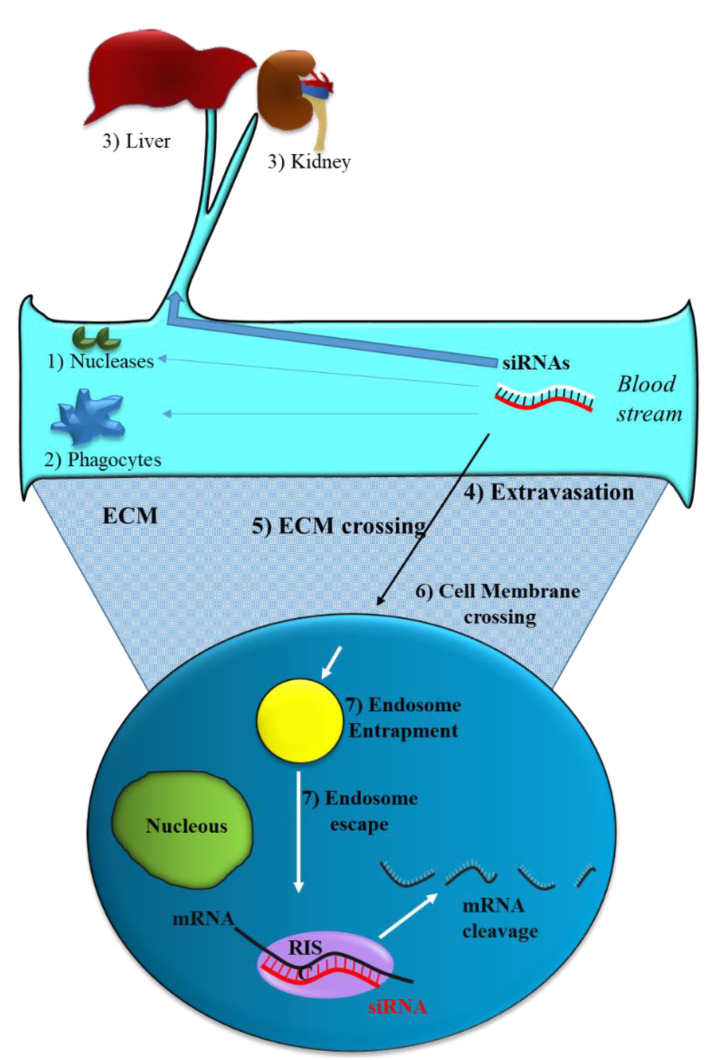
Obstacles to siRNA delivery. Systemically released siRNAs encounter blood nucleases (1), which can induce their rapid degradation together with the clearance by phagocytes (2) and by the liver–kidney sequestration/filtration (3). Extravasation (4), extra cellular matrix (ECM) crossing (5), cell membrane crossing (6) and endosomal escape (7) are the other barriers to be overcome by siRNAs.

**Figure 3 pharmaceutics-14-00718-f003:**
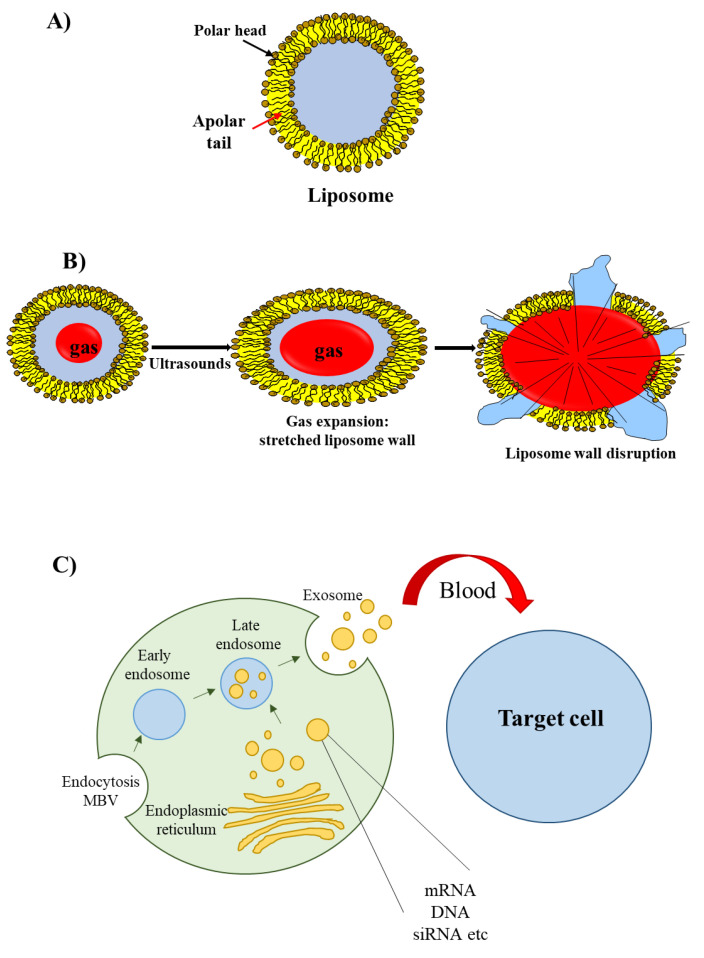
Liposomes, echogenic liposomes and exosomes. (**A**) The liposome wall consists of a double layer of amphiphilic lipids containing a polar head (oriented either towards the external or internal environments) and a polar tail (oriented towards the lipophilic region of the membrane). (**B**) Echogenic liposomes contain an inner gas phase that, in the presence of ultrasounds, undergoes expansion, contraction and vibration (cavitation); this phenomenon eventually determines liposome wall stretching and rupture. (**C**) Exosomes originate from early and late endosomes, which derive from invagination of the limited multi-vesicular body membrane (MBV); during this process, cellular proteins, mRNA, miRNA and DNA fragment are incorporated within the exosome; following exosome delivery form the mother cell, cellular proteins/mRNA/miRNA/DNA fragment are released to recipient cells.

**Figure 4 pharmaceutics-14-00718-f004:**
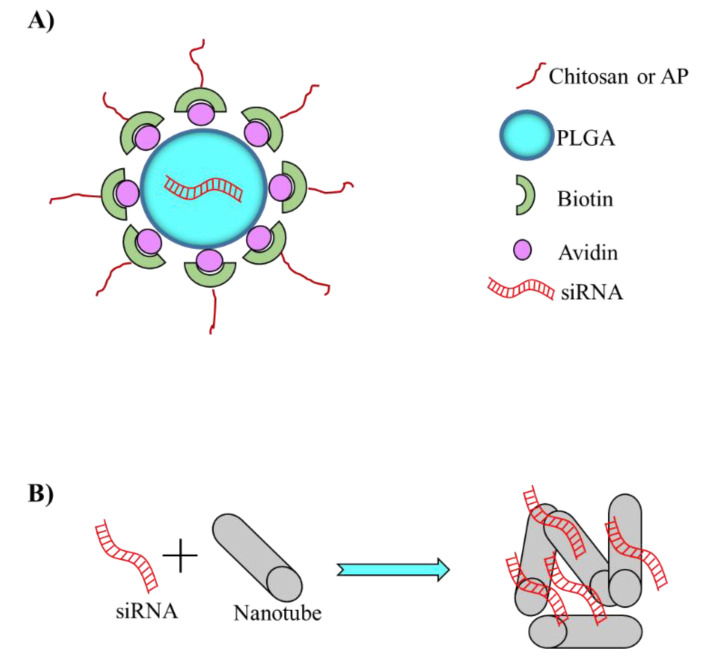
(**A**) PLGA nanoparticles were modified by the addition of either chitosan or penetratin (AP) via biotin-avidin link, from [72]. (**B**) Fe-doped chrysotile nanotubes were developed to carry and deliver siRNA, from [76].

**Figure 5 pharmaceutics-14-00718-f005:**
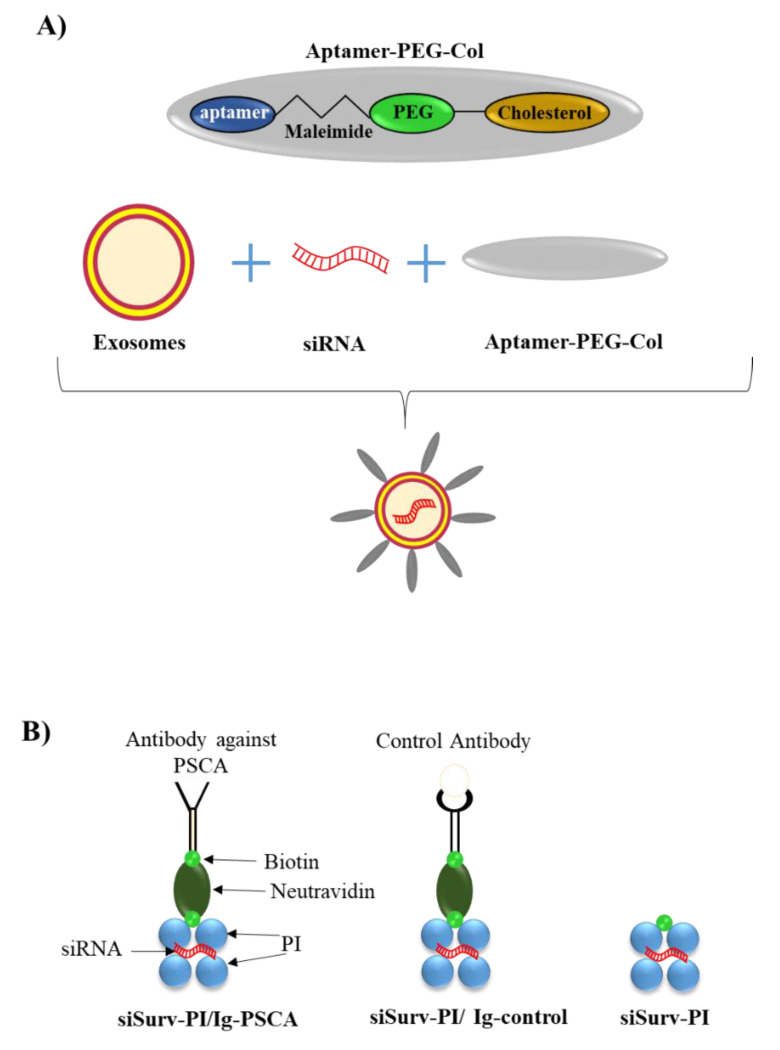
SiRNA delivery complexes in PC. (**A**) Exosome surface was modified by the addiction of AptE3 linked to maleimide-PEG-cholesterol molecules, from [91]. (**B**) Delivery system developed in [94]: It contains maltose-modified poly(propylene imine) polymers (PI) linked to biotin and loaded with siSurv (siSurv-PI). siSurv-PI was bound to an anti PSCA antibody linked to biotin (Ig-PSCA); siSurv-PI and Ig-PSCA were combined using a multi-biotin binding element (neutravidin).

**Table 1 pharmaceutics-14-00718-t001:** Properties of siRNA delivery materials.

Delivery Material	Advantages	Disadvantages
Liposomes	Easy siRNA loading Minor toxicity Structure easily tunable	No targeting specificity unless equipped with targeting moieties
Echogenic liposomes	As above with the possibility to induce ultrasound controlled delivery	Not applicable in deep tissue
Exosomes	As above with excellent biodistribution and the possibility to escape clearance by the mononuclear phagocyte system	No targeting specificity
Polymers	Production/isolation non expensive Structure easily tunable In general non toxic Possibility to escape endosome (PEI) In general easy siRNA loading Targeting ability to CD44 (HA)	Described toxicity for PEI Low solubility for CH Electrostatic repulsion of siRNA due to polyanionic nature for HA
Aptamers	Non toxic Able to target any desired molecule For DNA aptamers low production cost Can easily be stored for very long periods without losing their activity	RNA aptamers may be unstable in the biological environment. The selection procedure may be complex
Magnetic nanoparticles	Large surface area which can be functionalized with smart functional groups Magnetic behavior allows targeting to a defined tisse following application of an external magnetic field	Need functional groups on their surface for siRNA loading
Carbon nanotubes	High drug loading capacity Cellular uptake can be modulated varying the dimension	Poorly biodegradable

PEI: polyethylenimine; CH: chitosan; CD44: cluster determinant 44; HA: Hyaluronic acid.

## Data Availability

Not applicable.

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
