# Peer review of "An Overview of siRNA Delivery Strategies for Urological Cancers"

_pharmaceutics, 2022, doi:10.3390/pharmaceutics14040718_

Round 1

Reviewer 1 Report

Manuscript Number: Pharmaceutics-1603983

Manuscript: An overview of siRNA delivery strategies for urological cancers

Additional Comments:

RNA therapeutics is a matured drug development platform with multiple drugs in commercial use and many more are in the advance stages of clinical trial.  Approaches to improve the delivery of RNA therapeutics to specific tissue/cells type is necessary for fully realize the potential of this drug discovery platform. In this review, authors reviewed multiple delivery approaches reported for improving delivery of siRNA to develop drugs to treat different urological cancers. All the recent works in this area of research is cited in this article. In my opinion, the article is suitable for publication in Pharmaceutics after minor revision.

  1. Authors could improve the quality of figures in this manuscript.

Author Response

1) RNA therapeutics is a matured drug development platform with multiple drugs in commercial use and many more are in the advance stages of clinical trial.  Approaches to improve the delivery of RNA therapeutics to specific tissue/cells type is necessary for fully realize the potential of this drug discovery platform. In this review, authors reviewed multiple delivery approaches reported for improving delivery of siRNA to develop drugs to treat different urological cancers. All the recent works in this area of research is cited in this article. In my opinion, the article is suitable for publication in Pharmaceutics after minor revision.

We wish to thank the reviewer for the appreciation of our work

2) Authors could improve the quality of figures in this manuscript.

For the figures, we have used the classical power point tool and used it at the best of our possibility; moreover, we tried to keep the figures as simple as possible to guarantee the full comprehension also by non-expert in the field.

Reviewer 2 Report

The present review discusses various aspects regarding delivery of small interfering RNA for the purpose of preventing or restraining the urological cancers. Herein, the authors have assembled characteristics of location specific cancerous developments and related mechanisms. In this context, they have compared constitution of delivery materials used as vendors to send the siRNA species to the affected area. Herein, the authors have mentioned the essence of targeted delivery of siRNA without affecting much by the enzymes and the antibodies. The present form of this manuscript seems to have covered substantial portion of the subject matter. However, the manuscript seems to be incomplete in the following aspects:

  1. The manuscript though referred previous review works in the similar field, however, they should compare and highlight the topic(s) that was not covered by the earlier or contemporary review works (doi: 10.3390/nano7040077; doi: 10.3390/bioengineering7030091; doi: 10.1016/j.carbpol.2021.117809).

  1. Authors have mentioned various kinds of delivery materials suitable for sending siRNA species to desirable location or target. It would have been more interesting to the readers, if the authors make a table assembling merits and demerits of those delivery materials in delivering the siRNA to the affected cells or tissues.

  1. Often development of cancerous cells is associated with pH changes in the affected area. Moreover, drug delivery is closely associated with pH changes at different parts of our body. Accordingly, authors include a topic discussing pH dependent swelling/ deswelling and release of siRNA by various materials including liposomes.

  1. Authors should have included roles of plasmids and transposons in delivering siRNA species.

  1. Additionally, there are some minor errors that should be rectified. For instance, the first sentence of the introduction, ‘Urological cancer term is referring to cancer, which includes bladder, prostate and renal and cancers’ seems to be incorrect.

As a whole, the present form of the manuscript needs some necessary modifications to make it ready for further processing.

Author Response

The present review discusses various aspects regarding delivery of small interfering RNA for the purpose of preventing or restraining the urological cancers. Herein, the authors have assembled characteristics of location specific cancerous developments and related mechanisms. In this context, they have compared constitution of delivery materials used as vendors to send the siRNA species to the affected area. Herein, the authors have mentioned the essence of targeted delivery of siRNA without affecting much by the enzymes and the antibodies. The present form of this manuscript seems to have covered substantial portion of the subject matter. However, the manuscript seems to be incomplete in the following aspects:

1-The manuscript though referred previous review works in the similar field, however, they should compare and highlight the topic(s) that was not covered by the earlier or contemporary review works (doi: 10.3390/nano7040077; doi: 10.3390/bioengineering7030091; doi: 10.1016/j.carbpol.2021.117809).

In the revised manuscript, we have acknowledged the excellent papers mentioned on lines 102-105.

2-Authors have mentioned various kinds of delivery materials suitable for sending siRNA species to desirable location or target. It would have been more interesting to the readers, if the authors make a table assembling merits and demerits of those delivery materials in delivering the siRNA to the affected cells or tissues.

In the revised manuscript, we have introduced the novel table 1, which summarizes merits and demerits of the delivery materials presented; in addition, also in response to point 1 of reviewer 3, section 2.2 has been cut down to 38% of the original length.

3-Often development of cancerous cells is associated with pH changes in the affected area. Moreover, drug delivery is closely associated with pH changes at different parts of our body. Accordingly, authors include a topic discussing pH dependent swelling/ deswelling and release of siRNA by various materials including liposomes.

In the revised manuscript, in section 2.1 (lines149-158), section 2.3 (lines 87-193) and in section   2.4 (lines 233-244) we have discussed the pH decrease in tumor tissue, the generation of pH responsive liposomes and pH responsive polymers, respectively.

4-Authors should have included roles of plasmids and transposons in delivering siRNA species .

As we have not included papers based on these delivery approaches, we feel it is better not to discuss them in the present review.

5-Additionally, there are some minor errors that should be rectified. For instance, the first sentence of the introduction, ‘Urological cancer term is referring to cancer, which includes bladder, prostate and renal and cancers’ seems to be incorrect.

In the revised manuscript, the sentence has been correct (line 33); moreover, the entire manuscript has been checked out.

6-As a whole, the present form of the manuscript needs some necessary modifications to make it ready for further processing.

See the above modifications together with those prepared to answer the point raised by reviewer 3

Reviewer 3 Report

     This manuscript by Halib et al summarizes various siRNA-based delivery systems for treating urological cancers. In particular, the authors focused their discussion on the bladder cancer (BC), prostate cancer (PC) and renal cancer (RC). This manuscript is structured by firstly introducing general characteristics of various carrier systems and following by a discussion on specific examples related to urological cancers (BC, PC and RC). It should be mentioned that more discussions are centered on describing the general characteristics of the carrier systems and only few specific examples (siRNA-based systems) are considered towards treating urological cancers. Overall, this manuscript is well written. However, the authors should take into consideration on the following points to make this manuscript more helpful to the readers.

Specific comments:

  1. As highlighted in the title, the main objective of this manuscript is to provide an overview of the siRNA-based strategies for the treatment of urological cancers. However, a great deal of attention has been paid to the discussion on the characteristics of general carrier systems such lipid, polymers, iron oxide and carbon nanoparticles (Sections 2). Since these discussions are rudimentary and well known, the authors can omit/make it short or summarize briefly in a table.
  1. The authors have reviewed only a few specific examples in the sections 3.2. and 3.3, i.e. siRNA delivery in PC and RC. For example, only one example each is provided in sections 3.2.2 and 3.2.3. This manuscript can be strengthened by providing more specific examples (more recent updated references) in these fields.
  2. Similarly, this manuscript can be strengthened by including a section discussing the challenges and opportunities of combinatorial therapeutic approaches with RNAi and anticancer drugs/other treatment modalities.
  1. Besides the need to develop appropriate carrier systems, please highlight real practical challenges in the clinical settings of treating urological cancers.
  2. Again, the authors can revise the conclusion section by highlighting important limitations and future directions.
  3. In page no. 7, line no. 222, the sentence "polyethylene glycol (PEG) can be used to bind specific ligand that are fixed on the surface of delivery particles" is confusing. Please elaborate how only PEG can be used to bind specific ligands?
  4. Please check the repeating residue of chitosan in page no 7 line no 235.
  5. Spell error: Page no. 12, line no. 469, and page 16 line no 587, “fluorescent die” should be corrected as “fluorescent dye”.

Author Response

This manuscript by Halib et al summarizes various siRNA-based delivery systems for treating urological cancers. In particular, the authors focused their discussion on the bladder cancer (BC), prostate cancer (PC) and renal cancer (RC). This manuscript is structured by firstly introducing general characteristics of various carrier systems and following by a discussion on specific examples related to urological cancers (BC, PC and RC). It should be mentioned that more discussions are centered on describing the general characteristics of the carrier systems and only few specific examples (siRNA-based systems) are considered towards treating urological cancers. Overall, this manuscript is well written. However, the authors should take into consideration on the following points to make this manuscript more helpful to the readers.

Specific comments:

1) As highlighted in the title, the main objective of this manuscript is to provide an overview of the siRNA-based strategies for the treatment of urological cancers. However, a great deal of attention has been paid to the discussion on the characteristics of general carrier systems such lipid, polymers, iron oxide and carbon nanoparticles (Sections 2). Since these discussions are rudimentary and well known, the authors can omit/make it short or summarize briefly in a table.

In the revised manuscript, section 2.2 has been cut down to 38% of the original length; moreover, to answer point 2 of reviewer 2 we have introduced the novel table 1, which summarizes merits and demerits of the delivery materials presented.

2) The authors have reviewed only a few specific examples in the sections 3.2. and 3.3, i.e. siRNA delivery in PC and RC. For example, only one example each is provided in sections 3.2.2  and 3.2.3 (one added). This manuscript can be strengthened by providing more specific examples (more recent updated references) in these fields.

In the revised manuscript, we have introduced five novel recent papers in section 3.2.1 (lines 495-568 and lines 632-644), two novel papers in section 3.2.2 (lines 679-735), one in section 3.2.3 (lines 758-779) and one in section 3.3.1 (lines 805-821).  Moreover, we have specified in lines 101-102 that the review is focused “..on recently published papers without however, omitting some noticeable works of the past 5-10 years”

3) Similarly, this manuscript can be strengthened by including a section discussing the challenges and opportunities of combinatorial therapeutic approaches with RNAi and anticancer drugs/other treatment modalities.

This is an interesting topic which we have not systematically discussed in the present review as our focus was the use of siRNA alone; despite this, different examples of a combined siRNA/anticancer drug are reported (see refs 68, 71, 87, 91, 114 and 125).

4) Besides the need to develop appropriate carrier systems, please highlight real practical challenges in the clinical settings of treating urological cancers.

In the revised manuscript, we have highlighted the challenges related to the current available therapies for urological cancers (see lines 78-81, 83-88, 92-100)

5) Again, the authors can revise the conclusion section by highlighting important limitations and future directions.

In the revised manuscript, the conclusions have been re-written (see line 893-922) following the reviewer indications.

6) In page no. 7, line no. 222, the sentence "polyethylene glycol (PEG) can be used to bind specific ligand that are fixed on the surface of delivery particles" is confusing. Please elaborate how only PEG can be used to bind specific ligands?

In the revised manuscript, this sentence has been cut due to the shortening of section 2.4

7) Please check the repeating residue of chitosan in page no 7 line no 235.

We do not fully understand what the reviewer means;  we just point out that the part about CH has been reduced due to the shortening of section 2.4

8) Spell error: Page no. 12, line no. 469, and page 16 line no 587, “fluorescent die” should be corrected as “fluorescent dye”.

In the revised manuscript, the spell errors have been corrected

Round 2

Reviewer 2 Report

The authors have considered most of the issues raised in my earlier review, and accordingly this paper maybe accepted for publication